# The Impacts of Establishing Pilot National Parks on Local Residents' Livelihoods and Their Coping Strategies in China: A Case Study of Qilianshan National Park

**Jian Peng** [1,*], **Honglin Xiao** [2], **Rui Wang** [1] and **Yuanyuan Qi** [1]

1. School of Management, Minzu University of China, Beijing 100081, China; wryyzrh@163.com (R.W.); valerieqi@163.com (Y.Q.)
2. Department of History and Geography, Elon University, Elon, NC 27244, USA; hxiao@elon.edu
* Correspondence: pengkarst75@aliyun.com

**Abstract:** National Parks are a category of protected areas that emphasizes the sustainable use of park resources. China is a latecomer regarding the establishment of a national park system. In 2013, the Chinese administrative authorities announced the establishment of its own national park system to better protect the country's natural heritage and the integrity of its large ecosystems. Since 2015, ten pilot national parks have been designated to explore a pathway to implement the national park system better. Local communities are among the most critical stakeholders in establishing and managing a national park. Park management wouldn't be successful without the local residents' support and active involvement. Since national parks are positioned in China as protected areas with the highest priority to nature protection, their impacts on the local people, either inhabiting the park or living nearby, are unprecedented in the country. The park–people relationship is not a new topic in national park research; however, in the context of China's social, economic, and political framework, very little is known about the livelihood impacts of establishing a national park on the local residents and what strategies those residents should adapt to cope. In this study, the authors attempt to reveal the livelihood impacts of the national park creation in China on the local residents and their adaptation approaches by taking northwestern China's Qilianshan National Park (QNP) as a case. The study results indicate that the establishment of QNP and its policies have significantly impacted the livelihoods of the local residents. The more they rely on the park resources, the greater the magnitude of the effect has been, whether they reside inside or outside the park. Overall, the negative livelihood impacts exceed the positive ones at present. Still, most of the local residents wish to sustain their current livelihoods if the park's impacts can be made more manageable for them. Rarely do the local residents try to find alternative livelihoods unless they absolutely cannot overcome the negative impacts caused by the park's policies.

**Keywords:** national park; livelihood impact; livelihood strategy; community; Qilianshan; China

## 1. Introduction

National Parks (NP) are a category of protected areas regarded as a successful and sustainable model for managing and using natural resources [1]. It prioritizes protection while emphasizing the necessary and rational use of natural resources [2]. For this reason, many countries have established their own national park systems over the past century. China is a latecomer in terms of establishing a national park system. The goal to create its own national park system was not explicitly integrated into China's strategic development plan until the Third Plenary Session of the 18th Central Committee of the Communist Party of China (CPC) in 2013. Soon after, ten pilot national parks were designated with an aim to explore best practices for national park implementation. According to the Overall Plan for Establishing a National Park System (shortened to the Overall Plan in the following text) jointly issued in 2017 by the General Offices of the State Councils and the CPC's Central

Committee, national parks are defined in China as a category of protected areas, in which the highest priority is to protect nature.

Previous research indicates that it can be very challenging to manage a national park successfully without the active participation and support of the local community [3–7]. Although the Overall Plan states that national parks in China should be installed in conjunction with other functions, such as scientific research, environmental education, and public recreation, the Chinese central government emphasizes the implementation of the strictest policies to protect the park's resources and ecosystems. As in many other developing countries, a number of people reside in or near the designated pilot national parks. Most of these residents are heavily dependent on the natural resources in the parks to make their living. Disputes and conflicts concerning park protection vs. community development may be unprecedentedly intense in these situations. It is critical that park managers in China understand how park policies may impact the local residents' livelihoods and how to gain their support for park conservation.

The national park idea was born in North America; however, national park systems adopted and practiced by other countries are not the same as that of the USA [8]. On the contrary, they are very diverse due to varying institutional arrangements for politics, economies, and cultural traditions in different countries [8–13]. China is very different from other countries in terms of its political and social system, as well as cultural traditions; therefore, it will inevitably take a unique approach to establish its own national park system. It is not realistic for China to attempt to replicate the experiences of countries with mature national park systems [1,14]. The local community is one of the most crucial stakeholders and plays an essential role in establishing a national park [15]. On the one hand, local residents make their livings to a great extent by using park resources, and their livelihoods are heavily influenced by the policies created for the national parks. On the other hand, their attitudes toward and support for the national parks can significantly affect the achievement of park management goals.

Qilianshan National Park (QNP) is one of the ten pilot national parks in China, designated by the Chinese government in 2017. It is estimated that more than 50,000 people are living in or near the park, and a majority of them depend on the park resources to make their living. Conflicts concerning nature protection vs. resource use are common there, as is typical in all ten pilot national parks. In this article, we attempt to address the following questions by examining QNP as a case study: (1) What policies have the park administration made regarding this pilot park? (2) What are the impacts of these policies on the local residents' livelihoods? Furthermore, (3) how do the local people respond to these impacts, and what strategies can they adopt to more successfully cope with these impacts?

As the establishment of the national park system is new in China, very little is known there about these questions, especially in the specific context of China. Using the QNP case study, we intend to show how park policies may influence the local residents' livelihoods in all the ten pilot national parks of China and suggest general coping strategies. We will also propose suggestions to help the relevant Chinese park authorities to handle the residents' livelihoods issues better. Firstly, we present a literature review of research regarding park–people issues and national park studies within China. Then, we describe the study area, research methodology, and data collection process that we adopted. Based on the field survey in QNP, we discuss specific impacts on the local residents' livelihoods that were caused by the park's policies and strategies they adopted to cope. Then, we compare our research findings to existing relevant research and finally propose management suggestions for park authorities in QNP as well as other pilot park administrators in China.

## 2. Literature Review

### 2.1. Research on Park–Community Issues

Achieving a harmonious relationship between park protection and community development is a worldwide challenge for national park policymakers and management

authorities [16,17]. There are two opposing viewpoints regarding the proper attitude of park managers when dealing with local people. The first states that the primary purpose of establishing a national park is to protect the wilderness within it and human beings are not supposed to live inside [18,19]. From this perspective, a national park is a protected area designated to serve the public's recreational demands, and so, private interests ought not to interfere with the public goal [20]. If managers are influenced by this philosophy, local people tend to be viewed by the park authorities as a significant threat to nature protection; therefore, they advocate the adoption of the fortress-and-fine model to protect the park's resources [21]. The model prescribes the creation of special management units in which park rangers or guards are employed to patrol [22–24]. However, this management model ignores the rights and demands of the local people for survival and development, and it often causes disastrous consequences to their livelihoods [15]. As a result, local residents usually resist this model, and it becomes difficult to achieve the planned conservation goals due to ensuing social conflicts [15,17,25,26]. The other viewpoint argues that the local community should be treated as an inseparable part of a national park and that it is difficult to achieve the national park management goals without the support and involvement of the local people [17]. The International Union for Conservation of Nature (IUCN) emphasizes that nature protection is not the only goal of a national park; promoting the development of the local community should also be included [2]. Over the past few decades, three aspects of this model have been well addressed concerning park–people issues.

The first is the impact of the national park policies and management on the local community. Many developing countries adopt the fortress-and-fine model to manage national parks. Indigenous people are usually resettled and prohibited from engaging in private productive activities there. Consequently, their traditional livelihoods are heavily constrained [24,27–30]. In these cases, human–animal conflicts within or near national parks usually intensify. In many countries, large wild animals roam beyond the park's border and sometimes attack human beings, property, and livestock. This frequently causes enormous property and life losses among local residents. For example, tigers and leopards frequently attack and kill livestock and even people in Nepal's Bardia National Park [31]; in Mozambique's Limpopo National Park, elephants often walk out of the park and destroy crops [32]. In addition to these livelihood impacts, a national park may also entail other perceptible social and cultural impacts for local people, such as weakened human–land connections, constraints on religious rituals, increasing numbers of social conflicts, and poor community cohesion [24,33]. These problems are mainly associated with fence-and-fine park policies and regulations, which exclude local people from living and collecting materials in the national park and thus result in increased competition for limited natural resources.

The second aspect is the Indigenous residents' attitudes toward national parks. Existing research indicates that local residents usually support the necessity to protect the natural ecosystem and resources within national parks [29]. However, they can become very unsatisfied with and resistant to the park policies, which exclude them and ignore their substantial demands regarding their survival and development needs [34]. They may be forced to use park resources illegally to maintain their livelihoods through fishing, hunting, lumbering, and collecting. Sometimes, physical confrontations occur between local people and park rangers or guards [15,28,29]. Local residents only tend to hold positive attitudes toward park policies that bring them more benefits than costs [23,35,36]. Some studies reveal that attitudes of local residents toward national parks are heterogeneous and vary across residence locations, gender, income ranges, ages, and educational levels [37–41].

The third aspect is how to balance the park–people relationship. This is a worldwide challenge for park administrative authorities, as it is difficult to eliminate all conflicts between nature protection and community development [16]. Measures to reduce conflicts and earn the support of local residents is one of the hot topics in the research concerning park–people issues [17]. Some researchers argue that park managers should view the local residents as opportunities and partners rather than as threats and opponents and aban-

don authoritarianism and the fortress-and-fine management philosophy to alternatively adopt a model of co-management or integrated conservation [23,28]. Meanwhile, park managers should take active measures to reverse the local residents' negative attitudes and gather their support for the park protection policies [6,30]. For example, with further environmental education among local residents, they can better understand the meaning and significance of national parks [6,20,29]. Many researchers call for a consultative mechanism to reduce conflicts and increase collaboration among the relevant park stakeholders [20,42]. Other researchers advocate empowering the local residents by encouraging them to become involved in national park planning and policymaking [3,20]. In some countries, park managers compensate residents for any losses directly caused by park policies, or they provide subsidies for the livelihoods with low environmental and ecological impacts [43,44]. Ecological tourism might be the most frequently promoted strategy to minimize conflicts in national parks because it can create many alternative job opportunities [35,45]. To alleviate the park–people tensions and conflicts, some park managers make the zoning policy more flexible by setting up buffer zones between the park and residential areas for locals [46,47]. Some studies demonstrate that fair benefits distribution with a sensible assignment of obligations can positively reduce social conflicts and help achieve the park management goals [5,45]. Some researchers state that a national park should not be merely an enclave set aside for nature protection [17]. Rather, it should be an indispensable part of the human ecosystem. Park–community issues must therefore be studied using an ecological approach [15,42]. Ferreira [47] further argued that a symbiotic relationship should be developed between the national park and the local community so that both sides can benefit.

### 2.2. National Park Research in China

Although the Chinese government did not officially develop its own national park system until 2013, Chinese researchers have discussed national parks since the early 1980s. The first journal paper on national parks in China was published in 1980. Overall, national park studies by Chinese researchers can be divided into three stages. The first stage lasted for 25 years, from 1980 to 2005 (see Figure 1). National Parks were not a popular research topic during this period, and fewer than ten peer-reviewed journal articles were published on it each year. These limited discussions mainly focused on the concept and history of national parks and their practical implementations, particularly in North America [48,49].

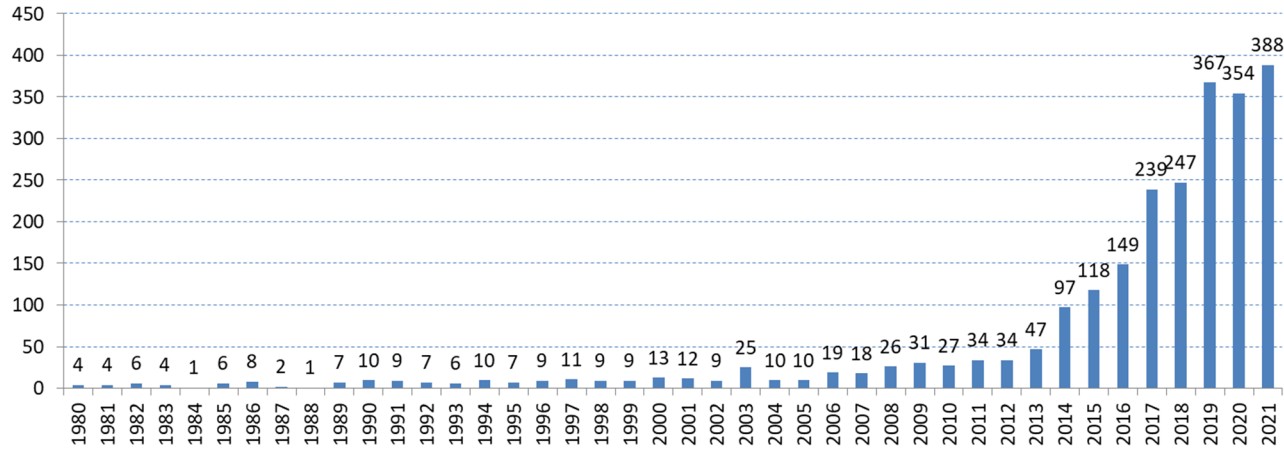

**Figure 1.** Annual publications regarding national park research in the Chinese scholarly journals.

In the late 1990s, Yunnan Province in southwestern China, with the help of The Nature Conservancy (TNC), began to explore how to put the idea of national parks into practice. In 2006, Potatso National Park, the first protected area named a national park in China, was established in the province. Thereafter, several other national parks like Potatso were

designated by the Yunnan provincial government. Although the central government of China did not formally recognize these parks, they were deemed the first practical national park in the country. Gradually, the concepts became more influential and attracted more attention from the Chinese academic community. National park research in China then entered its second stage (see Figure 1), which lasted for around six years until 2012. This period was characterized by an increase in studies on national parks. Related annual publications in the Chinese journals during this period varied from 20 to 40. Their research themes also became more diverse. With the hope to successfully create a government-sponsored national park system in China, successful administration experiences from other countries were heavily studied [9,10,50–56].

After the Chinese government declared the establishment of its national park system in 2013, 'National Park' immediately became a buzzword within the Chinese academic community, and national park research in China entered its third stage (see Figure 1). The average number of annual journal papers published from 2013 to 2021 exceeded 200. Compared with the previous stage, many more Chinese researchers discussed and studied national park-related topics. Consequently, the amount of national park literature (including journal articles and books) produced in this period increased significantly. Most of those publications were in Chinese, with very few appearing in English journals [57–59]. The research themes additionally grew much more diverse. After the central government of China decided to establish its own national park system, there was an urgent need for China to learn from international experiences. Many researchers continued to study, in more detail, the management experiences of national parks in other countries, especially in developed nations such as the US, Canada, the UK, France, New Zealand, Australia, and Japan [11,60–64]. Numerous studies have also been aimed at the prospect of establishing a unique national park system with Chinese characteristics [65–74]. Other studies have been conducted on the challenges confronting the pilot national parks [71,75,76]. Although National Park is a category of protected area that prioritizes protection, considering the integral connection between national parks and tourism, the potential benefits and rationality for tourism development in China's national parks also have been discussed extensively [77–80]. Since the local community is one of the most important stakeholders when establishing a national park [81], some researchers have argued that the relationship between National Parks and the local community is inseparable [14]. Thus, many studies focus on the mechanism to most productively promote the development of local, national park communities [13,75,82,83].

## 3. Methodology and Data Collection

### 3.1. Study Area

The Qilian Mountain is one of the famous high mountains in China, with an average altitude beyond 3000 m above sea level. It stretches from northwest to southeast and also is a border mountain between the Tibetan Plateau (usually high and cold) and the Gansu Corridor (usually low and dry). Thus, the Qilian Mountain area is rich in biodiversity and unique in terms of its alpine ecosystem. In 2015, it was recognized by the Ministry of Ecology and Environment of China as one of the 32 regions with priority to biodiversity protection in the country. It is not only a rarely-seen pool for alpine germplasms but also an important migration corridor for alpine wild animals. According to a scientific survey, the Qilian Mountain is home to a number of endangered flora and fauna, such as snow leopards, wild yaks, Equus kiangs, white-lipped deer, red deer, blue sheep as well as Chinese caterpillar fungi and Saussuree.

To protect the fragile alpine ecosystem and the endangered wild species inhabiting there, various Chinese governmental departments have established a number of protected areas over the past decades, such as nature reserves, forest parks, wetland parks, and geological parks. These protected areas are usually overlapped, and their administration units are often in conflict. Consequently, the state of nature protection is not satisfying in this region. For example, although mining in a nature reserve is illegal in China, there

were around 144 mining sites in Qilianshan National Reserve, which gave rise to severe environmental consequences (such as deforestation and soil erosion) as well as alarming damages to the natural ecosystem in this region. It is just in consideration of its outstanding significance in biology and ecology that a pilot national park is established in this area. In 2017, to thoroughly overcome the internal weakness of the past administration models and to make better and more effective protection of its fragile alpine ecosystem, QNP was designated as one of the ten pilot national parks by the central government of China. It lies in northwestern China's Gansu Province and Qinghai Province (see Figure 2), which covers an area of around 50,200 square kilometers. It is composed of two parts, i.e., the Gansu section (34,400 square kilometers) and the Qinghai section (15,800 square kilometers). Before it became a pilot national park, there had been eight different protected areas co-existing in this region, including three National Nature Reserves, one National Forest Park, three Provincial Forest Parks, and one National Wetland Park. All of them were created and administrated by different governments.

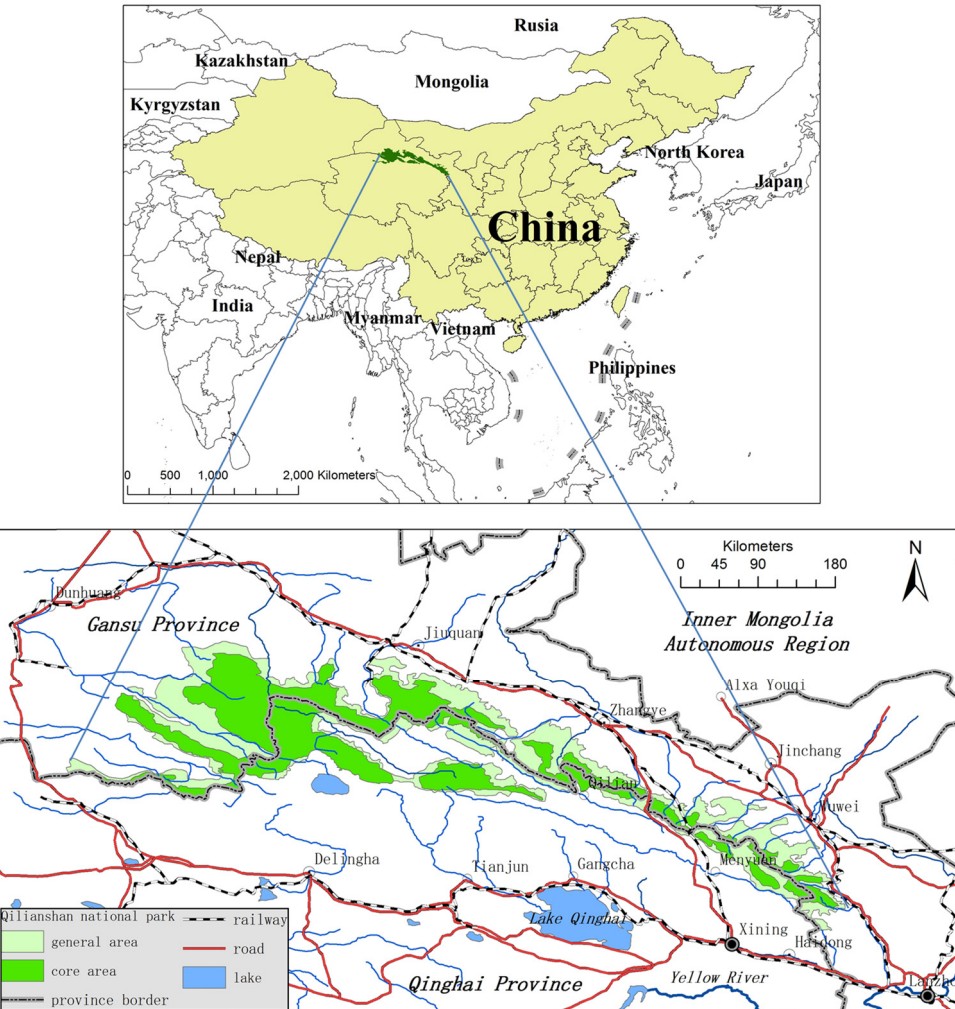

**Figure 2.** Location and management zoning of Qilianshan National Park departments. In 2017, they were merged into a bigger protected area, i.e., QNP, to better protect the endangered species and the fragile ecosystem in the Qilian Mountains area, where alpine meadows and forests primarily occupy the middle and lower parts, and glaciers cover the upper part.

Unlike the US's national parks, tens of thousands of residents have lived in all the pilot national parks for generations in China. According to surveys by the local governments, around 54,665 people are now living within QNP. Most of the local residents there make their living in agriculture-related jobs, such as animal grazing and crop planting. Though

QNP is a huge pilot national park covering more than 50,000 square kilometers, occupying territory in two different provinces, and encompassing several counties and dozens of towns and villages, there is no statistical data present concerning local residents' livelihoods at the park level. However, the statistical data at the village level does reflect the livelihoods of local residents, as well as the way they use the park's resources. For example, Guomi is one of the numerous villages in QNP, and the lives of its villagers are representative in the park. According to the data provided by the village head, this village covers an area of around 7611 hectares and is home to 175 households and 515 villagers. Some 79.6% of the land in Guomi is covered by grassland. As a result, most of its villagers make their livings by grazing livestock, mainly cattle and sheep. Grazing is a relatively vulnerable livelihood that is influenced by various factors, such as climate, workforce, markets, etc. Household incomes earned from grazing are usually not stable. In Guomi, 17 households or 53 people are recognized as financially "poor" by the local government. Most poor people are herdsmen whose livelihoods heavily depend on the grazing in the park.

According to the master plan for the park, QNP is zoned into two sections to better balance nature protection and community development, i.e., the core protected area and the general control area. The strictest protection policies are carried out in the core protected area, including relocating Indigenous residents outside of the park. Human activities, such as grazing, farming, mining, and industrial development, are constrained in the general control area; however, activities without too many negative impacts, such as scientific research, experiencing nature (tourism), monitoring, and environmental education, are permitted. As QNP covers a vast area, for this study, it is difficult to examine every part of the park due to time, funding, and workforce limitations. Therefore, we selected Qilian County, which lies in the heartland of Mt. Qilian, as the specific study area to examine the impacts of QNP's policies on the local residents' livelihoods.

*3.2. Methodology*

The national park system is still in a formative stage in China, and little is known about its impacts on the local residents' livelihoods. In consideration of this, we adopt in this study an emic and bottom-up approach to address the livelihood impacts of establishing a national park in the Mt. Qilian area. The impacts exerted by QNP were surveyed by interviewing the local residents, particularly those farmers and herdsmen whose livings heavily rely on the park's resources. In addition, certain other critical people, such as local government officials, park managers, and village heads, were also interviewed. As a well-proven social research method initially originated in the field of communication, Content Analysis(CA) is applied extensively in other social studies [84]. In this study, CA was employed to analyze the data collected from the in-depth interviews. A systematic coding and categorizing method was used to label the large amounts of textual information and to determine trends and patterns of the words used, their frequencies, their relationships, and the structures and discourses of the communications [85].

The Life History Method was adapted for the design of the interview questions. A life history is a person's narrated life story. However, it is different from other narrative stories due to its connection with real life or social events [86]. When examined in a particular social, historical, economic, and political context, a life story becomes meaningful to researchers [87]. The impacts of a given event on a person can be deeply studied using the Life History Method. The establishment of QNP was undoubtedly a big event for the residents living there, which produced numerous impacts on their livelihoods. With the Life History Method, these livelihood impacts can be identified by focusing on the changes before and after the park's creation.

*3.3. The Data Collection*

The most important part of the data collection is the interviews with the local residents of QNP, particularly those who live inside or near the park and whose livelihood heavily depends on the park resources. They are one of the key stakeholders in the establishment

of QNP and are prone to be greatly influenced by the park creation policies. To fully understand how the park policies affect their livelihoods and how they cope with the impacts, a set of carefully-designed questions was delivered to all the villagers interviewed. These included: (1) How did you and your family make your livings before the establishment of QNP? (2) Have there been any changes in your livelihoods since QNP became a pilot national park? What are they, specifically? Furthermore, (3) What strategies did your family adopt to cope with any negative impacts from QNP on your livelihoods?

The data collection was conducted in July 2019 and lasted for around 15 days. Considering most of the local residents are ethnic minority people (mainly Tibetan), all the interviews were done at the home of the respondents with the help of a local Tibetan guide. In total, 68 villagers from different households were surveyed (see Table 1). They are from five villages (namely Guomi, Bianma, Hedong, Qingyanggou, and Rixu) of three different towns (namely Zhamashi, Yeniugou, and Arou). Specifically, 37 of them are from Guomi, 12 are from Bianma, 3 are from Hedong, 14 are from Qingyanggou, and 2 are from Rixu. Among these, 49 lived inside the park and the other 19 outside; 57 households earned their incomes mainly by grazing or farming; most of the respondents were poorly educated (see Table 2). There were 30.8% of them that have never gone to school, 41.2% have an elementary school education, and only 28% receive a high school education or above. All the interviews were recorded with the agreement of the respondent and transcribed into texts for Content Analysis.

**Table 1.** Description of the interviewed respondents living in and near Qilianshan National Park.

| Village Location | Village Name | Households | Livelihood |
| --- | --- | --- | --- |
| Inside the park | Guomi | 30 | Farming/grazing |
| | | 7 | Non-farming/grazing |
| | Bianma | 12 | Farming/grazing |
| | Hedong | 3 | Farming/grazing |
| Outside (but close to) the park | Qingyanggou | 12 | Farming/grazing |
| | | 2 | Non-farming/grazing |
| | Rixu | 2 | Non-farming/grazing |

**Table 2.** Demographic profile of the respondents.

| Profile | Classification | Frequency | Percentage (%) |
| --- | --- | --- | --- |
| Gender | Male | 62 | 91.2 |
| | Female | 6 | 8.8 |
| Age | Under 20 | 2 | 2.9 |
| | 20–40 | 33 | 48.5 |
| | 40–60 | 28 | 41.2 |
| | Above 60 | 5 | 7.4 |
| Ethnicity | Han | 8 | 11.7 |
| | Tibetan | 50 | 73.5 |
| | Mongol | 5 | 7.4 |
| | Hui | 5 | 7.4 |
| Education | Never attended school | 21 | 30.8 |
| | Elementary school | 28 | 41.2 |
| | Junior middle school | 14 | 20.6 |
| | Senior middle school | 5 | 7.4 |
| | Some college | 0 | 0 |
| | Undergraduate and above | 0 | 0 |

Before starting our field survey in these villages, we visited the Qinghai branch of QNP Administration in Xining, the capital city of Qinghai Province, and interviewed a senior park manager to have a general understanding of the pilot park establishment schedule and progress (particularly what policies they had made and implemented). Then,

we went to Qilian County and visited some relevant governmental agencies, including Culture, Radio, Television, and Tourism Administration as well as Forestry and Grassland Administration. Some officials from these agencies were interviewed to further understand how the park policies were carried out at the county level and what difficulties they were confronted with.

## 4. The Impacts of QNP on the Local Residents' Livelihoods

The in-depth interviews with the local officials and park managers indicate that a series of policies were announced by the QNP park authorities and the local governments to promote its establishment. However, when this field investigation was conducted in 2019, only part of those policies had been put into practice, including no-grazing in the core protected area, closure of the mines, suspension of construction projects, closure or suspension of tourist attractions, and employing local villagers affected by the park creation as ecological guards. Our field investigation in Qilian County indicated that these policies had apparent impacts on the local residents' livelihoods to differing extents.

### 4.1. The Livelihood Impacts of the No-Grazing Policy

According to the QNP master plan, human activities are prohibited in the core protected area to protect the fragile alpine ecosystem better. It is explicitly stated that grazing will not be allowed in the core protected area anymore, and the Indigenous people living there will be resettled out of the park sometime in the future. When this field investigation was conducted in July 2019, the resettlement policy was not yet implemented. However, the no-grazing policy was already being enforced in part of the core protected area (mainly in Ebao Town, where the shared summer pastureland of the Qingyanggou villagers lay). This policy has imposed noticeable impacts on the livelihoods of the local herders, who rely mainly on the pastureland that they own in the park to make their livings. A direct consequence caused by this policy was that the pastureland available for the villagers was significantly reduced, especially the summer pastureland. In the past, the Qingyanggou villagers could graze their herds in Ebao for about two months in the summer. After the no-grazing policy was enforced in the core protected area of QNP in 2018, they were not allowed to graze there anymore, and the grazing time was shortened to one month due to the shortage of grass. As a result, some villagers could not raise livestock as many as they did in the past, and their income from grazing decreased enormously. Some villagers had to buy more fodder to feed their livestock. Although they received some monetary compensation from the government, it was far less than the losses resulting from this reduction of the pasturelands and consequently their livestock.

Nima, a villager from Qingyanggou, said that too many people are now grazing in the summer pastureland. He found that the grass seemed insufficient to feed the livestock even though they had just moved to the summer pastureland no more than 20 days before. After the establishment of QNP, his family had to graze fewer herds than before, reducing them by about 20 cattle and 200 sheep. Although the local government compensated each household with 250 kg of fodder for their loss of summer pastureland in the core protected area, that was far less than needed to feed the herds. Duorijiefucaidan, another villager from Qingyanggou, said his family received compensation consisting of ten bags of fodder from the local government. However, a half bag is needed each day, and the fodder they receive will be consumed very soon. Some villagers explicitly stated that their family income had very noticeably decreased since the implementation of the no-grazing policy. Duorijiedongzhibu, a villager also from Qingyanggou, said that the government's compensation was much less than their loss, only about 10,000 to 20,000 yuan per year.

In this study, 57 of the 68 villagers interviewed stated that they were apprehensive about not being allowed to graze anymore after the creation of QNP. Some villagers did not make a living by grazing, but they owned pastureland in the park. They usually rented out their pasture to other herders who wanted to graze more livestock. When the no-grazing policy was implemented, these villagers might lose their rentals and pastureland. Some

villagers earned extra income by collecting caterpillar fungi or grassland mushrooms, sold at reasonable prices. When they were prohibited from grazing, this income also was reduced. Table 3 shows that 64 of the 68 villagers interviewed offered their worries about the possible loss of their pasturelands in the future, while 57 had experienced a reduction of herds sizes, 19 experienced income losses from being unable now to collect the caterpillar fungi, and 5 from grassland mushrooms. Additionally, many herders in the Mt. Qilian area use cow dung as a household fuel, which can help reduce their fuel costs to some extent. Two villagers of Guomi Village explicitly mentioned that they were worried that they no longer would be able to collect cow dung as much quantity as they had before and consequently would have to pay more to buy extra household fuel (such as coal) because of the no-grazing policy.

**Table 3.** Perceived impacts of park establishment by respondents in this survey.

| Perceived Livelihood Impacts | Frequency | Percentage (%) |
| --- | --- | --- |
| Reduced pastureland | 64 | 94.12 |
| Less income from grazing | 57 | 83.82 |
| Loss of income from renting out pastureland | 10 | 14.71 |
| Less cow dung fuel | 57 | 83.82 |
| Less income from collecting caterpillar fungus | 19 | 27.94 |
| Less income from collecting grassland mushroom | 5 | 7.35 |
| Less tourism income | 8 | 11.76 |
| Fewer tourism jobs | 6 | 8.82 |
| Fewer jobs and income from construction projects | 8 | 11.76 |
| More jobs and extra income from being an ecological ranger | 9 | 13.24 |

### 4.2. The Livelihood Impacts of Closing the Mines and Suspending the Construction Projects

The Mt. Qilian area is rich in mineral resources and is called in Chinese "Wanbaoshan," which means literally "Ten Thousand Treasures Mountain." According to a geological survey, this area is home to nearly 15 metal and 20 non-metal minerals. Over the past several decades, many mineral enterprises have swarmed into this area to mine these resources, creating job opportunities for the local residents. According to a China Central Television (CCTV) report in 2017, as many as 144 mining sites existed in the Qilianshan National Nature Reserve. Some residents made their living from working in these mining sites. By Chinese law, mining in nature reserves is illegal. With the designation of Qilianshan as a pilot national park in 2017, all these unlawful mining sites were immediately closed. Consequently, these residents lost their mining jobs and had to find a substitutive livelihood. Content Analysis revealed that 8 of the 68 interviewees mentioned they had worked in the mining sites of the Mt. Qilian area and could earn several thousand *yuan* per month (see Table 3). When these mining sites were closed, they lost their jobs and incomes soon after. Pengmaodanzhou, a villager from Guomi, said he used to work as a truck driver at a mining site in QNP for several years, in which he could earn an income of 6000 yuan per month. He lost this job soon after QNP was designated a pilot national park, which resulted in the closure of that mining site. In addition, there were several ongoing projects to build the infrastructure, mining, or tourism facilities, which had also provided the local residents many job opportunities—the pay for doing construction work varied from 100 to 200 yuan per day. Many of them were suspended after QNP was designated a pilot national park in 2017. Among the 68 villagers interviewed, 8 respondents mentioned that they had worked in these construction projects but lost their jobs soon after the announcement that QNP had become a pilot national park.

### 4.3. The Livelihood Impacts of Closing or Suspending the Tourist Attractions

Mt. Qilian is well known in China for its gorgeous alpine landscape. It attracts numerous tourists each year. A majority of them are self-driving tourists from nearby cities or provinces. According to data issued by the Tourism Bureau of Qilian County, the

county received approximately 4.5 million visitors from 2016 to 2018, which generated 11,800 tourism-related jobs and tourism revenue of around 17.68 billion yuan. In 2019, before the COVID-19 pandemic, the county received approximately 2.8 million tourists. The successful development of the tourism industry has created many job opportunities. Quite a few of the residents living in or near QNP made their living or earned extra money by providing tourists with needed services in the food, accommodations, transportation, guide, performance, and recreation services (horse-riding and archery were local recreation activities). According to our content analysis, 17 of the 68 villagers interviewed have worked in tourism-related jobs, with 5 running tourism-related businesses (such as a homestay, restaurant, hotel, or grocery), and 12 working in tourism reception services (such as horse-riding, cooking, and cleaning).

Tourism is usually regarded in China as a significant threat to nature protection. After 2017 at QNP, the well-being of the natural areas dropped significantly. The park authorities decided to close or suspend some tourist attractions and scenic spots to reduce the negative impacts of the tourism activities. For example, there is a mountain glacier (called *Ba Yi Bing Chuan* in Chinese) in the west Qilian County. Many tourists from the nearby provinces (Sichuan, Gansu, Xinjiang, Ningxia, etc.) would drive hundreds of miles to visit it. This glacier was advertised prominently. However, it was closed in 2017 due to the establishment of QNP. The local people witnessed a perceivable decrease in visitors to Qilian County, and consequently, this impacted those whose livelihoods relied on tourism.

After losing his job as a truck driver due to the closing of a mining site, Pengmaodanzhou, the villager aforementioned from Guomi, ran a family hotel with his wife on the side of the road leading from Qilian County to the Ba Yi glacier. He said that the number of self-driving tourists who passed the village had decreased. His tourism income had also decreased significantly; even in the summertime, very few tourists visit. In addition, other residents lost their tourism-related jobs, and their incomes also suffered severely. There were 14 of the 68 villagers interviewed who stated that tourism-related job opportunities had apparently been reduced due to the implementation of the park policies (see Table 3). Pengmaosuoan, a villager from Hedong (a village close to Guomi), had been employed as a cook at a family hotel run by his cousin; he earned a monthly income of around 8000 yuan. Due to the closing of the Ba Yi glacier and the dwindling of tourists, it is increasingly challenging for the family hotel to survive. Most likely, he said he would lose this job in the near future.

*4.4. The Livelihood Impacts of the Ecological Guards Policy*

To better protect the fragile alpine ecosystem in QNP, a policy for the creation of ecological guards (*Sheng Tai Xun Hu Yuan* in Chinese) was implemented in 2018. This policy is intended to create job opportunities, to alleviate the negative impacts of the no-grazing policy on the livelihoods of the local villagers. Each guard is typically paid by the government 1700 yuan per month or 20,400 yuan per year. The park administered the policy to patrol and report any potential danger or threat (such as fire and illegal hunting) to the park authorities. The guards are also responsible for cleaning litter in the park. Specifically for the program, members of households that have been affected by the park establishment should be eligible to apply for these jobs. However, only one member from each family could have the job. At present, there are 15 ecological guards in Guomi Village and more than 200 in Zhamashi Town. In this study, 9 of the 68 villagers interviewed were employed as ecological guards (see Table 3). They stated that they did benefit from this policy, which offset to some extent the potential economic losses caused by the no-grazing and other conservation policies. However, their pay still was usually less than what the villagers could earn from grazing. Many respondents were worried that they would face financial hardship if the no-grazing policy were fully implemented because they could not find alternative sources of income.

## 5. Responses of the Local Residents

The above analysis shows that QNP has already generated various impacts on the livelihoods of the local residents, particularly those villagers who depend on the natural resources in the park. To more fully distinguish the responses of the local people, we divided them into two groups. The first group includes villagers whose livelihoods heavily depended on the natural resources in the park (particularly pastureland and farmland) in the park and who generated income mainly from grazing or farming. The second group contains villagers who do not primarily earn income from the natural resources in the park. Then, we discussed in detail the strategies they used to cope with regard to each group separately (see Table 4).

**Table 4.** Strategies adopted by the respondents to cope with the park's policies.

| (Potential) Strategies | Frequency | Percentage (%) |
|---|---|---|
| Giving up grazing and looking for a construction job | 17 | 25.00 |
| Giving up grazing and taking a tourism job | 12 | 17.65 |
| Continuing grazing by renting other villagers' pastureland or buying more fodder | 4 | 5.88 |
| Continuing grazing by reducing herds size | 8 | 11.76 |
| Continuing running a restaurant by developing a local market | 2 | 2.94 |
| Looking for a local alternative job | 24 | 35.29 |
| Looking for a non-local alternative job | 21 | 30.88 |
| No strategies to deal with the no-grazing policy | 18 | 26.47 |

### 5.1. Responses of the Local People Living Mainly by Grazing and Farming

For the local residents who own land in the park and live primarily by grazing or farming, both the no-grazing and ecological guard policies have substantially impacted their livelihoods. There is a causal relationship between the no-grazing and ecological guard policies, livelihood impacts, and the adaptation responses of the local residents (see Figure 3). The no-grazing policy results in the reduction of their pastureland, which leads to a series of impacts on their income, including reduced grazing incomes, reduced incomes from collecting caterpillar fungi or grassland mushrooms, and also increased costs for household fuel. All those negative impacts cause them considerable financial difficulties, especially when the government's compensation for pastureland reduction is less than the losses that they have suffered. To cope with the no-grazing policy, these residents must devise alternative strategies. As a majority of the local villagers in QNP are poorly educated and do not have any other sources of income apart from grazing and farming, our field survey revealed that many of the respondents said the most common approach they took to preserve their original livelihoods was to graze fewer herds, rent pastureland from other villagers, breed herds in stables, or buy more fodder.

Eighteen out of the sixty-eight respondents stated that they did not know how to cope with the difficulties caused by the no-grazing policy. Another approach some herders took was to graze their herds in smaller pasturelands after losing some summer pastureland in the core protected area due to the no-grazing policy (see Figure 4). Other respondents mentioned adopting alternative strategies to maintain their living. For example, several respondents stated that if they could not graze due to the reduction of their pasturelands, they might give up grazing, sell their herds, and rent their pastures to those villagers who want to have more pastureland access. In general, the attitude of most of the villagers toward the no-grazing policy was passive. Overall, the local people living in and near QNP were positive and supportive of the ecological guard policy. It offers extra job opportunities to the villagers and increases their household incomes while providing nature conservation.

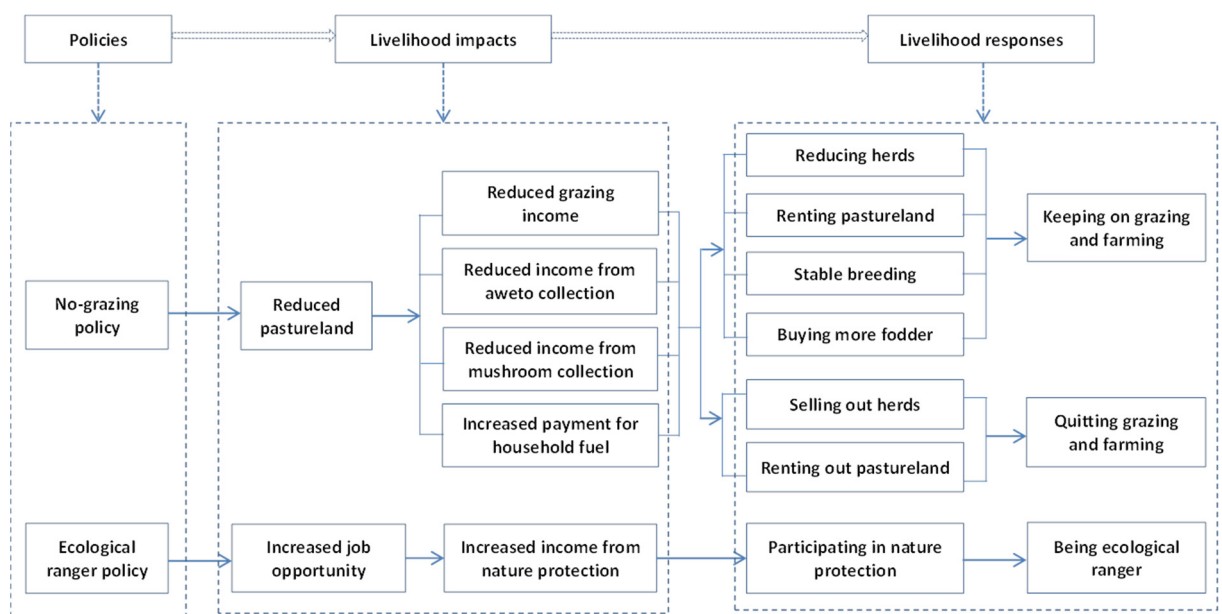

**Figure 3.** Livelihood impacts on the local residents due to the QNP establishment policies and their responses.

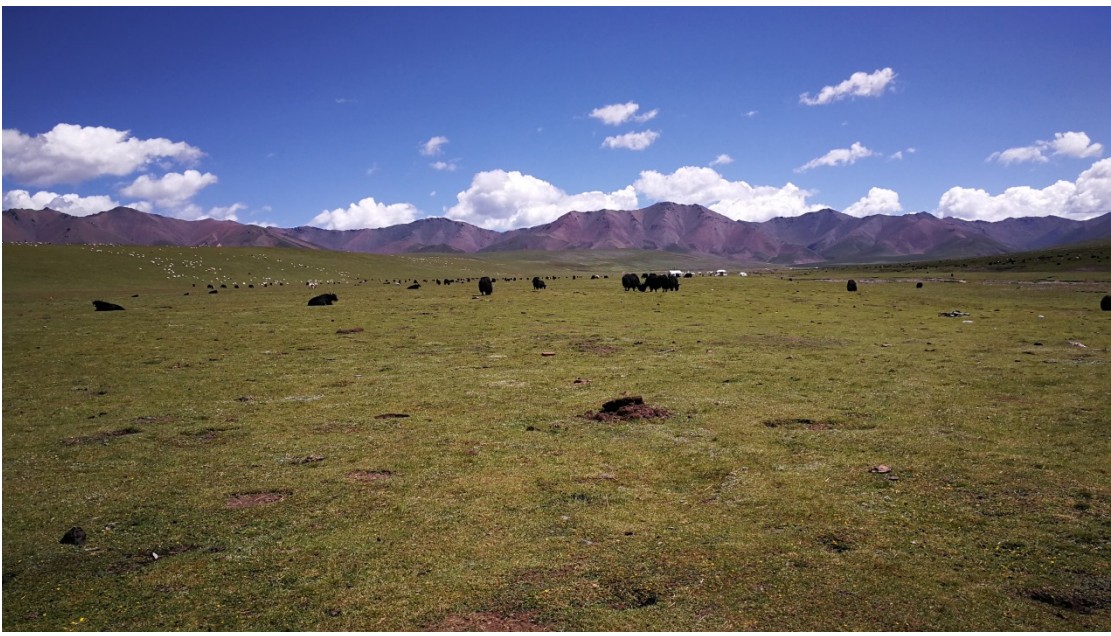

**Figure 4.** Summer pastureland crowded with more herds than before in Ebao.

*5.2. Responses of the Local People Not Earning Income Primarily by Grazing and Farming*

For the residents who live inside or near the park and do not make their livings primarily by grazing or farming, the impacts of QNP on their livelihoods are very different, as are their responses. Our study found that their livelihoods were influenced mainly by three policies, i.e., closure of the mining sites, suspension of the construction projects, and closure of the tourist attractions. Immediately after all the mining sites in the park were closed, many villagers lost their jobs and incomes. When construction projects were suspended, job opportunities were also reduced. Consequently, the villagers engaged in those occupations earned less income. Interestingly, the villagers influenced by these two policies adopted similar strategies to cope (see Figure 5). Some of them began to look for similar construction job opportunities elsewhere. Additionally, a few turned to tourism

to earn income by running a tourism-related business. Examples include converting their homes into homestays, as illustrated in Figure 6 (which features one of the newly opened homestays after its owner lost his job as a truck driver at a mining site). Relatively speaking, the villagers' responses to the closure of the tourist attractions and scenic spots were a little bit more complex. Facing reduced tourists numbers, some villagers who ran restaurants and homestays tried to stay in tourism by developing the local market or reducing their employees. A few abandoned tourism jobs and ran other businesses, such as grocery stores.

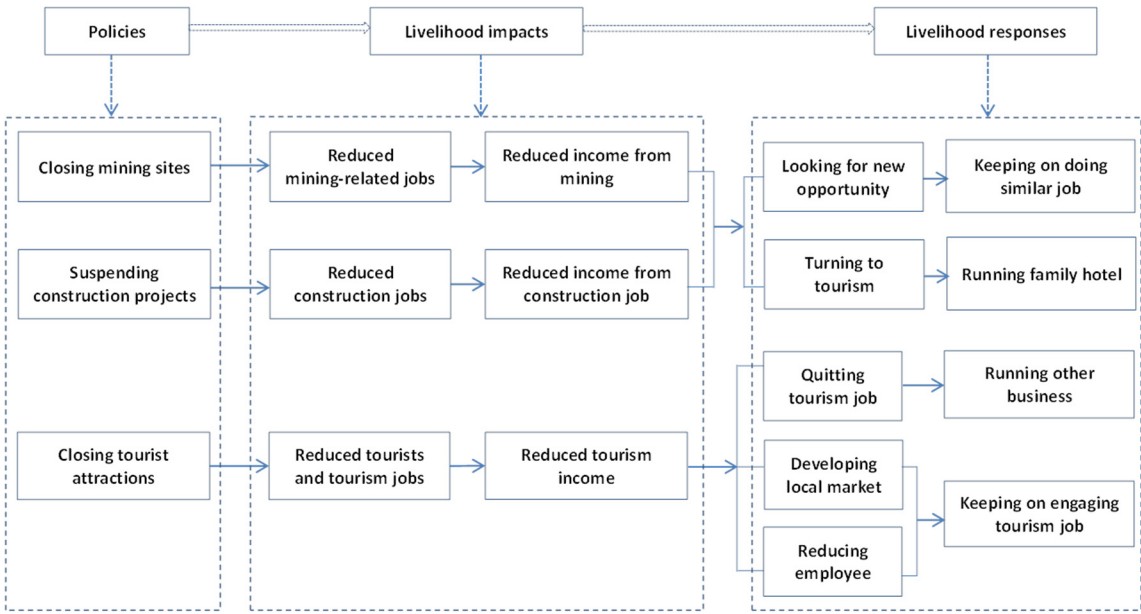

**Figure 5.** Livelihood for local residents due to the policies of Qilianshan National Park and their responses.

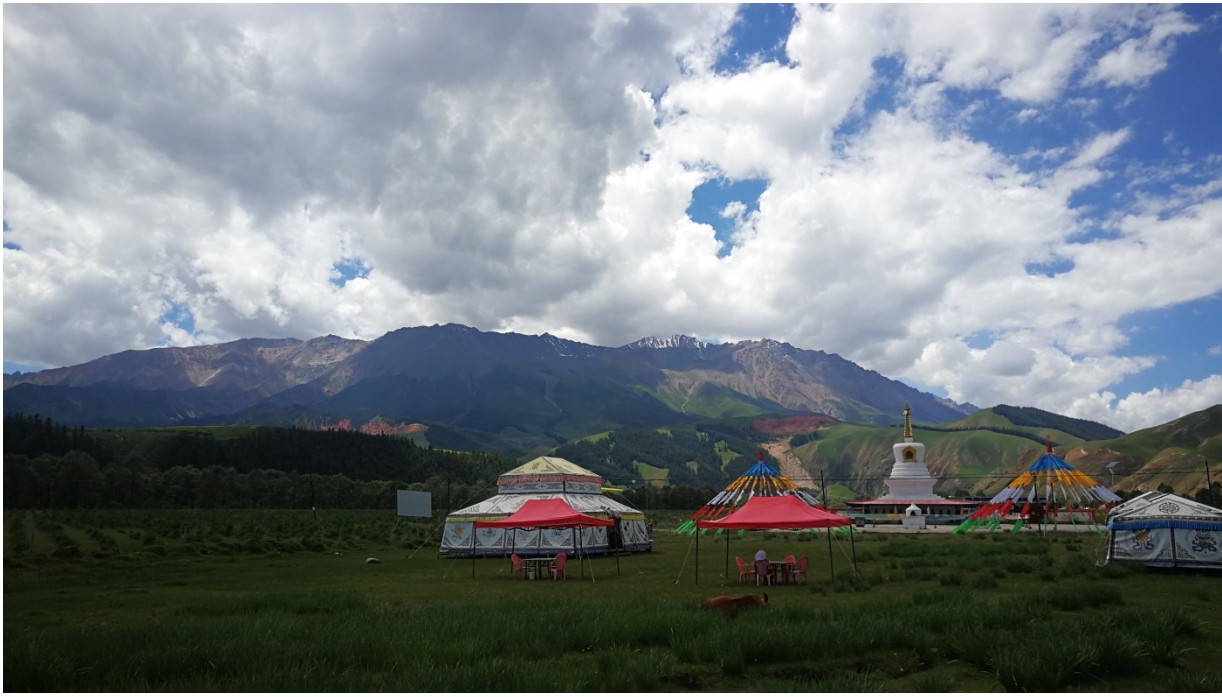

**Figure 6.** A Tibetan-style homestay newly opened by a villager in Guomi Village.

## 6. Discussion

Following the spread of the US national park idea to developing countries, the philosophy of fortress-and-fine conservation was extensively adopted to achieve stricter protection of nature [21]. Extant studies suggest that creating a national park usually brings forth negative impacts to the local residents in many ways but to different extents [24,27–30]. One of the most visible impacts is challenges to their traditional forms of livelihood due to restrictive park policies, particularly regarding access to park resources. Unlike the US, Canada, New Zealand, and Australia, national parks in many other countries are usually inhabited by many Indigenous people, who had lived there for generations before the NPs were created. A majority of these local people make their living by using the park resources. After the establishment of a national park, many locals living within or near it may lose access to these resources and therefore are confronted with livelihood problems [28,29]. In general, the more they depend on these natural resources, the more their livelihoods will be influenced by the creation of national parks.

We reached some similar findings during our empirical study in QNP. As in many other developing countries, the administrative authorities of China advocated the philosophy of strict protection when they created their own national park system. National Parks are defined in the country as a category of protected areas in which the overwhelming priority is nature protection. As a result, the fortress-and-fine conservation model is adopted to manage national parks. In QNP and the rest of the pilot national parks in China, guard stations have been established at entrances into the parks. According to the park regulations, ordinary visitors cannot enter the parks unless they have special permits from the park authorities. Although QNP is still in its formative stage of operation, impacts on local residents' livelihoods due to the park's policies have emerged. The natural resource on which the local people there depend the most is pastureland in the park. Livestock grazing is the most common use of land in QNP. The no-grazing policy heavily influences the herders' livelihoods by reducing their summer pasturelands. As a result, they cannot graze their livestock herds precisely as they had before, and they must spend more to maintain an equal number of livestock in their herds. Whether the herders live within or near the park, this policy significantly influences their livelihoods if their pastureland lies within the park. These villagers are poorly educated; therefore, it can be difficult for them to make a living by taking jobs other than grazing and farming. For those villagers who obtain their primary income from mining-related work, closures of the mining sites result in an immediate loss of their jobs. However, compared with the no-grazing policy, the financial impacts on the local community by mine closures are relatively small because only a small percentage of villagers are involved in the mining sector. According to our field investigation, only 8 out of the 68 respondents stated that they could not work anymore at the mining sites due to the establishment of QNP. Most of these respondents have some skills, such as driving or building; compared with the other herders and farmers, their immediate job restrictions were more manageable because they could sometimes find alternative jobs outside of the park.

In addition to nature conservation, meeting the recreational demands of the public is another critical function of a national park. Tourism is believed to be the best way to realize this function [15]. Often, tourism is advocated by the park authorities and is supported by the local people because it can create a variety of job opportunities to solve the livelihood issue caused by the restrictive park policies. The park authorities can utilize tourism as a strategy or tool to solve local park–people conflicts [15,88]. However, experts in nature conservation and park authorities in China also view tourism as a threat to nature protection. As a point of fact, tourism is rarely mentioned in the Overall Plan for Establishing a National Park System. In QNP, many tourist attractions were shut down soon after it was designated a pilot park, and this greatly limited the role of tourism to serve as an alternative livelihood for the local residents. Thus, the attitude of park authorities toward national park tourism represents one potential difference between China and other countries.

In 1982, the Third World Congress on National Parks was held in Bali, Indonesia. The relationship between park protection and human needs became the focal point of discussion for the first time [15]. After that, the management policies of national parks have emphasized the integration of park conservation with community development. The fortress conservation philosophy has been questioned and gradually replaced by co-management or participatory management involving local residents. Many Integrated Conservation and Development Programs (ICDPs) have been carried out worldwide [3]. Although the administrative authorities of China continued to give the top priority to nature protection and adopted the fortress conservation philosophy for the National Park management, they also began to emphasize the idea of green development, which is intended to balance nature conservation and community development [89]. In QNP, the ecological guard policy was implemented to provide conservation-related jobs for villagers who were influenced by the no-grazing policy. Nepal and Weber argue that benefit sharing is an effective strategy to alleviate park–people conflicts [15]. Employing local villagers to be ecological guards is not only beneficial to them but helps to earn their support for park conservation as well. However, considering that the local governments entirely pay the salaries of the ecological guards, the program might entail many financial burdens for local governments. In addition, this policy only covers a minimal number of locals whose livelihoods are influenced by the establishment of QNP. The guard policy can alleviate some park–people issues to a limited extent. Still, other measures are needed for the park managers to more fully and successfully address the alternative livelihood needs of many local residents in or near the park.

Although previous studies have extensively discussed local residents' perceptions of their livelihood impacts due to the creation of national parks [24,27–30], how they could productively respond to these impacts in different contexts is still poorly understood. After access to the park's resources is cut off for the local residents, most of them still desire to maintain their traditional livelihoods. This can lead them to enter the park to use the resources they need illegally. This response has been reported extensively by national park managers in the developing countries of Africa, Asia, and South America [6,20,24]. The situation is not exactly the same in China due to the country's unique political and economic system, as well as China's cultural traditions. Power is highly centralized in China, and collectivism is upheld by the Chinese people as a longstanding obligatory system; therefore, when a top-down policy that could lead to negative impacts on livelihoods is designed and implemented, most Chinese people usually choose to be compliant rather than openly resistant. Though such policies may negatively and inevitably affect them, most will prefer to bear the costs on their own and make adjustments to their original livelihoods. China is a socialist country, and the Chinese governments at both national and local levels regard the improvement of people's welfare and living conditions as a top priority. Over the past few decades, the Chinese government has attained widely recognized achievements in enhancing people's living quality and social equity. In this study, some villagers whom we interviewed explicitly stated their faith in the government and the park policies. They believe that the local governments will not make the public suffer from losses and that negative impacts caused by the park policies will be temporary. They also believe that finally, they can benefit more from the park's creation in the long term.

## 7. Conclusions and Implication

National Parks are a new category of protected areas in China, and establishing a national park system is a new job for the Chinese government. Since China is very different from other countries regarding the country's politics, economy, society, and traditional culture, the national park system in the country is fated to meet Chinese conditions. However, during the formative stage of China's National Park system, it is important and necessary to uncover how national parks are influencing the livelihoods of local people and how they are responding. The results of such studies can provide practical implications for National Park policymaking in China. In this study, by taking QNP as a case, the authors

conducted an empirical examination of the impacts on the livelihoods of local people and their corresponding responses. The findings from this study will be conducive to enriching our understanding of the complex effects of park policies in China.

Based on our field survey in QNP, the following conclusions can be drawn: (1) Compared with the definition and goals of national parks offered by IUCN, the National Park authorities in China overemphasize the protection of park resources and subsequently adopt the strictest protection policies. Several impacts of this method on local residents' livelihoods and financial well-being have emerged that should be paid enough attention to; (2) The QNP's policies have produced a variety of livelihood impacts. In general, adverse effects outnumbered the positive ones at the time that this study was conducted. The no-grazing policy, closures of mining sites, suspensions of construction projects, and tourist attractions closures have reduced natural assets, job opportunities, and family incomes. The ecological guard policy has created some conservation job opportunities and provided local residents some extra income, but only a limited number of villagers can benefit from this policy; (3) The livelihood impacts of establishing QNP on the local residents is heterogeneous. The more the local residents' livelihoods depend on the park resources, the more the park conservation policies affect their financial conditions. Local residents who earn their primary income from grazing suffer the most negative impacts; (4) Different strategies have been adopted by the local residents to cope with the livelihood challenges caused by the creation of QNP. Most have attempted to maintain their original livelihood, if possible. Others have sought alternative job opportunities when they could not handle the impacts.

When this study was conducted, the pilot establishment of QNP was not yet completed. According to the master plan for the park, 33 tasks were to be implemented before it could be recognized as a formal national park. Some problematic policies, such as resettlements and no-grazing, will be further implemented in the coming years. If the park authorities do not take active measures to solve the livelihood issues for local people, it is foreseeable that additional negative livelihood impacts will emerge. The park will be confronted with many social conflicts between the park management authorities and the local people, and the whole pilot efforts could potentially end as a social failure. Aside from the ecological guard policy, the park authorities currently have no other policies to create more alternative job opportunities for the local people. To earn their support and achieve the success of park creation, the park managers should and must try their best to provide alternative livelihoods for those residents who have been heavily influenced by the park's conservation policies. Policies regarding tourism should be revalued and adjusted since they can generate considerable job opportunities and lessen impairments to nature conservation, given proper administration and management. The park authorities should embrace a positive attitude toward tourism. Specific and responsible planning for tourism development is required in QNP and other pilot national parks in China. As some scholars argue [90], the worldwide conservation paradigm has gradually shifted to recognize the importance of a participatory and inclusive approach to protected area management; this has been implemented successfully in several places from the 1970s onwards. However, China continues to overemphasize strict protection of national parks, which will likely cause more avoidable troubles. If China does not properly handle the relationship between nature protection and resource use, it may take a much longer time to establish a national park system that meets the IUCN criterion.

Thus, the national park authorities in China need to make necessary adjustments to their current policies to promote their national park system better and earn the support of park residents. First, China's central government should not emphasize the strictest environmental protection in national parks; they must pay more attention to rational uses of park resources. According to the IUCN, National Parks are a category of protected areas that emphasizes both the protection of park resources and their reasonable use. If a national park is protected like a fortress and the local people's necessary uses of park resources are ignored, China will not have a real "national park." Secondly, the national park authorities in China should pay more attention to the livelihood issues of park residents

and understand that national parks cannot succeed without their support. They must adopt adequate measures to help solve the financial difficulties of local people caused by the park, such as providing monetary compensation as well as needed financial, knowledge, and policy support to help them successfully transition to alternative livelihoods. Thirdly, appropriate development of park tourism should be permitted and encouraged sustainably. As stated by the IUCN, meeting the public's recreational demands is one of the most basic functions of a national park. National parks must be open to all people and provide the public with recreational opportunities and services. Worldwide, no national park excludes visitors and tourism. Since tourism can generate many job opportunities, it also can play a crucial and positive role in helping local residents to find alternative livelihoods. In regions where the economy is underdeveloped, this is particularly significant. Of course, tourism development in the national parks of China must be ecologically sustainable. Careful planning is required to ensure that tourism does not impair the unique wild animals and ecosystems protected in its national parks.

**Author Contributions:** Conceptualization, J.P.; methodology, J.P.; software, J.P.; validation, J.P.; formal analysis, R.W.; investigation, J.P., R.W. and Y.Q.; resources, J.P.; data curation, R.W.; writing—original draft, J.P.; writing—review and editing, H.X.; visualization, J.P.; supervision, J.P.; funding acquisition, J.P. All authors have read and agreed to the published version of the manuscript.

**Funding:** This study is funded by the National Social Science Foundation of China (18BJY039) and the Beijing Longway Foundation (HX2019029).

**Institutional Review Board Statement:** Not applicable.

**Informed Consent Statement:** Not applicable.

**Data Availability Statement:** The data presented in this study are available on request from the corresponding author.

**Acknowledgments:** The authors would like to thank the Qilianshan National Park Administration officials and local villagers for their active support and participation in the field investigation. We are also grateful to the anonymous peer reviewers for their encouraging comments and pertinent suggestions. In addition, a special thanks is given to Lianyong Wang from China's Southwest University for his kind support for our research, and Qiushenji Zhuome for her excellent guide and interpretation job during our field investigation in Qilianshan National Park.

**Conflicts of Interest:** The authors declare no conflict of interest.

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
