# Peer review of "The Impacts of Establishing Pilot National Parks on Local Residents’ Livelihoods and Their Coping Strategies in China: A Case Study of Qilianshan National Park"

_sustainability, doi:10.3390/su14063537_

Round 1
Reviewer 1 Report
The MS deals with the institution of National Parks in China, with a park used as a model to assess livelihood impacts. I think it covers a very important gap of knowledge, particularly for China. Therefore, the MS can be a good paper, but I think it lacks some important parts, thus requiring only a few small revisions as following:
- The main issue is related to the language. Although I am not a native English speaker, the MS would benefit by a language polishing by a native English.
- The MS lacks aims and predictions, which should be added at the end of the introduction in a clear way, so to put the paper in an hypothesis-driven context.
- Figure 1 lacks unit of measurements and the “name” of neighboring countries.
Reviewer 2 Report
This is a thorough; most interesting paper that suits the scope of Sustainability journal that should be considered for publication. I here suggest changes to improve the draft:
- When the authors discuss the academic aspects of the development of National Parks in China, they could support their argument with a figure showing number of publications about this topic per year, country and / or language.
- When presenting the study site (Qilianshan NP), the authors should extend on a critical aspect of the establishment of the park: what are the ecological goals of creating the park? What is the state of the flora and fauna and its historical trajectory? Are there endangered species, and if so, what is their conservation status? Are problems such as deforestation and soil erosion and chronic problem in the area? It is vital to provide detail into the ‘ecological’ part of the story, as currently the article has too strong a focus on the human aspects and conflicts of the NP system, while not giving the readers information about what is being tried to protect in this region.
- To make the result section more synthetic, the authors could place the more descriptive passages (interview description) to supplementary information, and more briefly mention the main themes of each interview.
- The conclusion seems too long, repeating the abstract: I would suggest making it brief and simply highlighting the main results.
Reviewer 3 Report
Dear Authors, the article needs to be supplemented. Details are in the attachment. Information relating to Chinese politics should be supported by literature. The article describes in detail the case of one of the villagers, but did not refer to the information whether the establishment of the national park could have a positive impact on the local society. Describing one case in detail does not allow us to draw any clear conclusions. Other examples can be given. No proposals were made that could be introduced by the Chinese government.

Reviewer 4 Report
This paper is an interesting case study for evaluating potential impacts of establishing the pilot national park at Qilianshan, China. Authors used content analysis to reflect changes of local residents’livelihood. However, some questions have not been well addressed. Thus, my suggestion is to revise at this stage.
1. Method: In addition to the content analysis, Could you provide other quantitative indicators (e.g., statistical data) to reflect the residents’livelihood if possibile?
2. Some grammatical errors and irregular English writing can easily be found in the full text. Authors should seek a native speaker to improve them.
Round 2
Reviewer 3 Report
Changes in the text (corrections) resulted in a more precise presentation of the subject. I have no comments on the article. My comment No. 7 after the changes made is not important. In the future, the authors can make an additional analysis of what changes have occurred in the lives of people living inside the park and outside the park.